# Inverted U-Curve Association between Serum Indoxyl Sulfate Levels and Cardiovascular Events in Patients on Chronic Hemodialysis

**DOI:** 10.3390/jcm10040744

**Published:** 2021-02-13

**Authors:** Ming-Hsien Tsai, Chung-Hsin Chang, Hung-Hsiang Liou, Yu-Wei Fang

**Affiliations:** 1Division of Nephrology, Department of Internal Medicine, Shin Kong Wu Ho-Su Memorial Hospital, Taipei 11101, Taiwan; chaosmyth.tw@gmail.com (M.-H.T.); chunghsin0303@gmail.com (C.-H.C.); 2Department of Medicine, Fu-Jen Catholic University School of Medicine, New Taipei City 242062, Taiwan; 3Division of Nephrology, Department of Internal Medicine, Hsin-Jen Hospital, New Taipei City 24243, Taiwan; hh258527@ms23.hinet.net

**Keywords:** hemodialysis, indoxyl sulfate, p-cresyl sulfate, protein-bound uremic toxin, cardiovascular disease, mortality

## Abstract

Background: Protein-bound uremic toxins are associated with cardiovascular disease and mortality in patients with chronic kidney disease. We investigated their association with clinical outcomes in patients undergoing chronic hemodialysis (CHD). Methods: A prospective cohort study was conducted on 86 Taiwanese patients undergoing CHD. The predictors were indoxyl sulfate and p-cresyl sulfate concentrations, with each analyzed as three tertiles. Outcomes were cardiovascular events and all-cause mortality. Results: During a 25-month follow up period, there were 23 cardiovascular events and seven all-cause mortality events. In the crude survival analysis, the second indoxyl sulfate tertile was shown to be a powerful predictor of cardiovascular events compared with the third tertile (hazard ratio (HR), 3.14; 95% confidence interval (CI), 1.10–8.94), and the first tertile was shown to have a poor but insignificant cardiovascular outcome (HR, 1.09; 95% CI, 0.30–4.00). Moreover, the predictive power of the second indoxyl sulfate tertile for cardiovascular events remained after adjustment for confounders (HR, 5.42; 95% CI, 1.67–17.60). Conclusions: An inverse U-curve relationship was observed between the total serum indoxyl sulfate level and cardiovascular events in our CHD patients. A large-scale study is needed to confirm this relationship.

## 1. Introduction

Cardiovascular disease (CVD) is significantly more prevalent in patients on chronic hemodialysis (CHD) than in the general population because of the increased oxidative stress and inflammation, which contribute to vascular abnormalities that are associated with a higher death rate. Therefore, CVD is the main cause of morbidity and mortality among dialysis patients [1,2,3]. Protein-bound uremic toxins (PBUTs) have been implicated in uremic syndrome and may play an important role in the mortality and morbidity rates of patients with chronic kidney disease (CKD) [4,5]. In dialysis patients, such uremic toxins might have a significant impact on prognosis because of their low clearance rates by dialysis. Of these, indoxyl sulfate (IS) and p-cresyl sulfate (PCS) are perhaps the most widely studied marker molecules [6,7].

IS is derived from the dietary protein “tryptophan”, which is metabolized into indole by the intestinal flora, and indole is further converted to IS in the liver. IS plays a key role in CKD deterioration due to its effects on the progression of glomerular sclerosis and renal anemia, as well as the inhibition of endothelial proliferation [8,9]. IS seems to act as an endotheliotoxin leading a principal role on the pathogenesis of CVD in CKD [10]. Moreover, several studies have shown that IS is associated with atherosclerosis, cardiovascular events, and skeletal resistance to parathyroid hormone (PTH) in dialysis patients [11,12]. p-Cresol, an end-product of tyrosine metabolism in the gastrointestinal tract, exists predominantly as conjugated PCS in vivo; unconjugated p-cresol is not detectable [13]. It decreases the activation of polymorphonuclear granulocytes in a concentration-dependent manner, inhibits the release of platelet-activating factor by macrophages, and induces hepatotoxicity by increasing aluminum accumulation [14]. In addition, an increased level of serum-free p-cresol was found to be related to higher mortality and morbidity rates [15], as well as vascular access failure [16] in dialysis patients. These findings suggest the importance of PCS and IS in the clinical outcomes of dialysis patients.

However, previous studies have reported neutral findings regarding the relationship between PBUTs and clinical outcomes, including all-cause mortality and CV events in patients with CHD [17,18]. A reverse epidemiology finding linked to protein–energy wasting in dialysis patients has been reported [19,20]. Therefore, under the hypothesis that PBUTs might follow a reverse epidemiology pattern in dialysis patients, our study aimed to investigate the association pattern between serum levels of PCS and IS and clinical outcomes (new CV events and death) in CHD patients during a 25-month follow-up period.

## 2. Materials and Methods

### 2.1. Study Design and Population

This prospective observational cohort study recruited stable hemodialysis patients from June to August 2010 from a single medical center (Figure 1). Initially, 143 participants were screened. Patients with an acute infection, those who had suffered from cardiovascular events in the previous 3 months, those with malignancies or liver disease, those who were participating in drug clinical trials, those with serum albumin <2.9 g/dL, those <18 years or >80 years, and those without agreement were excluded from this study. All participants had been receiving 4 h of maintenance dialysis three times per week for a minimum of 3 months without reusing the dialyzer. Demographic data were obtained from the participants’ medical records upon entry into the study (baseline). Participants were excluded if they left our hemodialysis unit, shifted from hemodialysis to peritoneal dialysis, or received a kidney transplant. Finally, a total of 86 patients with stable CHD were recruited into our study. This study was performed in accordance with the principles of the Declaration of Helsinki and was approved by the Ethics Committee of the Shin-Kong Wu Ho-Su Memorial Hospital (20100608R). Informed consent was obtained from all participants.

### 2.2. Laboratory Data

All blood samples were obtained just before the dialysis procedure after 8 h of fasting, and the following variables were measured: blood urea nitrogen (BUN, mg/dL), creatinine (Cr, mg/dL), hemoglobin (Hb, g/dL), ionized calcium (iCa, mg/dL), phosphate (P, mg/dL), intact PTH (i-PTH, pg/mL), albumin (g/dL), highly sensitive C-reactive protein (hsCRP, mg/dL), homocysteine (μmol/L), total IS (mg/dL), total PCS (mg/dL), total cholesterol (TC, mg/dL), low-density lipoprotein (LDL, mg/dL), high-density lipoprotein (HDL, mg/dL), triglycerides (TGs, mg/dL), and dialysis adequacy (Kt/V). Total serum PCS and IS were analyzed using ultra-performance liquid chromatography twice per week (second and third session), and average values were calculated.

### 2.3. Event Evaluation

Participants were followed up until 30 September 2012. During the follow-up period, all-cause deaths and cardiovascular events, including death from cardiac causes, myocardial ischemia, cardiac arrhythmia, cerebrovascular accidents, or new onset of peripheral vascular disease, were recorded. To control the accuracy of the data, the chart notes were reviewed for all admissions. For each participant, the time to event was calculated as the time from the date of entry into the study until the date of the first studied events (mortality and CV), the date of quitting the study, or the end of the study, whichever came first.

### 2.4. Statistical Analysis

We express the data as the mean ± standard deviation (SD), range (interquartile range (IQR)) or frequency (percent), as appropriate. For analytical purposes, participants were divided according to IS tertiles and PCS tertiles. Intergroup comparisons were performed using the chi-square test for categorical variables and the Kruskal–Wallis test and Mann–Whitney U-test for continuous variables when the normal distribution assumption was violated. The Kaplan–Meyer method was used to estimate cardiovascular-event-free rates and overall survival for the IS and PCS tertiles. The log-rank test was used to compare the difference between survival curves. Moreover, univariate and multivariable analyses of cardiovascular events in different IS tertiles were performed using the Cox proportional hazards regression model under the assumption of proportional hazards, which was not violated by testing the interactions between time and variables. A modified stepwise procedure with five modeling steps was used for multivariable analyses. Moreover, a principal components analysis with inspection of the covariance matrix and a rotation method of varimax was performed for dimensionality reduction in parameters to avoid over-adjusting in the multivariable model. Finally, two principal components were generated, which were able to explain 99% of the variance. A two-sided *p* value of <0.05 was considered statistically significant, and *p* ≤ 0.018 was considered statistically significant in the post-hoc comparison, which was performed according to the Bonferroni method. Statistical analyses were conducted using the statistical package for social sciences (SPSS version 26; IBM Inc., Chicago, IL, USA).

### 2.5. Patient and Public Involvement

Patients and the public were not invited to comment on the study design or conduct of the study. However, they will be informed of the study results through publications.

## 3. Results

### 3.1. Study Population Characteristics

Eighty-six stable hemodialysis patients were enrolled in this prospective study. The mean age was 61.8 ± 12.4 years, the mean dialysis time was 9.1 ± 5.4 years, 61% (*n* = 53) were males, 42% (*n* = 36) were diabetic, and 30% (*n* = 26) had a history of CVD. The distributions of IS (mean, 34.2 mg/dL; SD, 13.8) and PCS (mean, 25.5 mg/dL; SD, 18.3) are shown in Figure 2. Table 1 shows the demographic and biochemical characteristics of the 86 included patients in the IS and PCS tertiles. A comparison of patients divided by IS tertiles revealed no significant intergroup differences with respect to age, gender, body mass index (BMI), diabetes mellitus status, systolic and diastolic arterial pressure values, lipid profile, calcium phosphate product, CRP, hemoglobin content, dialysis adequacy, HD vintage, or a positive history for cardiovascular events. However, i-PTH (*p* = 0.027), albumin (*p* = 0.012), and homocysteine (*p* = 0.011) concentrations differed significantly between IS tertiles. Serum levels of albumin were significantly higher in patients in the third IS tertile than in those in the first and second IS tertiles. Similarly, i-PTH levels were significantly higher in the third IS tertile than in the first tertile. Conversely, the homocysteine level was significantly higher for patients in the first IS tertile than in those in the third IS tertile. A comparison of the PCS tertiles revealed no significant intergroup differences with respect to age, gender, BMI, diabetes mellitus status, systolic and diastolic arterial pressures, HD vintage, or a positive history for cardiovascular events. No intergroup differences were noted with respect to biochemical characteristics (Table 1).

### 3.2. Cardiovascular Events

Twenty-three patients experienced a new CV event, with events including coronary artery disease (*n* = 11), peripheral artery occlusive disease (*n* = 1), stroke (*n* = 2), congestive heart failure (*n* = 1), intracranial hemorrhage (*n* = 2), atrial fibrillations (*n* = 1), and cardiovascular death (*n* = 5).

Table 2 shows that being in the second IS tertile (hazard ratio (HR): 3.14, 95% confidence interval (CI): 1.10–8.94, compared with the third tertile), having a history of previous CVD (HR, 2.64; 95%CI, 1.16–5.99), and having lower albumin (HR, 0.29, 95%CI, 0.11–0.75), greater hsCRP (HR, 1.26, 95%CI, 1.11–1.44), and lower LDL (HR, 0.98, 95%CI, 0.97–1.00) concentrations were associated with a greater likelihood of experiencing cardiovascular events in the crude analysis.

The Kaplan–Meyer analysis showed that the serum IS tertile was strongly associated with the rate of cardiovascular events, but PCS was not (χ^2^ = 6.14, *p* = 0.04 and χ^2^ = 3.99, *p* = 0.13, respectively) (Figure 3A,B). Table 3 shows the results of cumulative Cox model analyses of the effect of IS on cardiovascular events. Patients in the second IS tertile were found to be at a significantly increased risk of experiencing cardiovascular events. After adjustment for selected predictors of cardiovascular events (model 1: age, gender, and hemodialysis time; model 2: model 1 predictors plus previous CVD and DM; model 3: model 2 predictors plus homocysteine, calcium phosphate product, and hsCRP), patients in the second IS tertile were still found to have a significantly increased risk of experiencing cardiovascular events (models 1–3). In model 4 (model 3 predictors plus LDL and albumin), this association became insignificant.

In model 5 (model 3 predictors plus two principal components), where over-adjustment was considered, the association became significant. An inverted U-curve association between serum IS tertile and CV events was observed, showing that the second tertile had the worst CV outcome, followed by the third tertile. The first tertile had the best CV outcome.

### 3.3. All-Cause Mortality Events

Seven mortality events occurred. These were of cardiovascular (*n* = 5), infectious (*n* = 1), and other (*n* = 1) origin. All-cause mortality was shown to differ significantly with respect to serum albumin (HR, 0.05; 95% CI, 0.01–0.28), hsCRP (HR, 2.15; 95% CI, 1.41–3.30), uric acid (HR, 0.51; 95% CI, 0.28–0.90), and total cholesterol (HR, 0.97; 95% CI, 0.94–0.99) concentrations in the crude analysis. No significant associations were found among IS, PCS, and all-cause mortality (Table 2). Kaplan–Meyer analysis showed that the serum IS tertile and PCS were not significantly associated with all-cause death (χ^2^ = 0.38, *p* = 0.828 and χ^2^ = 0.25, *p* = 0.881, respectively) (Figure 4A,B).

## 4. Discussion

To the best of our knowledge, the present study is the first to demonstrate an inverted U-curve relationship between the risk of CV outcome and serum IS tertile in patients with CHD. Most importantly, we observed that having a moderate serum IS concentration was associated with an increased risk of experiencing a cardiovascular event. Patients with the highest and lowest levels of serum IS within the study cohort had lower levels of risk. After multi-variable adjustment, this association remained. The results of this study could provide physicians with more information about the clinical effects of PBUTs in hemodialysis patients.

Emerging evidence has shown that PBUTs may be directly responsible for the development of CVD in the patients with CKD [21]. Regarding the negative effect of PBUTs on the patients with CHD, some studies have reported serum-free PCS as a cardiovascular risk factor in non-diabetic hemodialysis patients [22], a predictor of infection-related hospitalization in hemodialysis patients [23], and a marker for all-cause mortality and cardiovascular disease in elderly hemodialysis patients [24]. Lin et al. reported that serum IS has no predictive power for CVD but that free p-cresol plays a significant role in CVD development in hemodialysis patients [23]. However, a post-hoc analysis of the HEMO trial, which recruited 1273 participants with a median observation time of 2.3 years, failed to demonstrate a linear association between serum levels of PCS and IS and cardiovascular outcomes in dialysis patients [18]. Moreover, a study by van Gelder MK et al. with 80 participants with dialysis who were followed for a median time of 4.3 years showed that the continuous forms of serum IS and PCS had no significant associations with all-cause mortality and CV events [17]. Therefore, the PBUTs seem to have no significantly linear association with CV events in dialysis patients.

Interestingly, our study shows an inverted U-curve association between serum IS concentration and CV outcome in patients with CHD when the IS concentrations were divided into tertiles. This finding can be explained by the phenomena of reverse epidemiology in dialysis patients due to the effect of nutritional status [19,20]. Patients’ nutrition might play an important role in this association, as a higher serum IS concentration indicates a better nutritional status, which is a strong predictor for CVD development in hemodialysis patients [25,26]. In our study, we observed that a higher serum IS level was significantly associated with a higher serum albumin level and with a lower homocysteine level, which can decrease the development of CVD [27,28]. Moreover, a positive trend between serum IS level and serum lipid level was also noted despite no statistical significance. Such findings in our study supported the positive association between serum IS level and nutrition status in the patients with CHD. Therefore, we hypothesized that the protective effect of better nutrition in hemodialysis patients overcomes the harmful consequences of having a serum IS concentration on the development of CVD. Moreover, this could explain why no significant linear associations between PBUTs and clinical outcomes in the patients with CHD were found in previous studies [17,18].

Notably, the oral charcoal adsorbent AST-120 (KREMEZIN^®^) can adsorb hydrophobic uremic toxins, such as IS and p-cresol, in the gastrointestinal tract and then decrease their levels in the serum [29,30]. In clinical studies, administration of AST-120 was reported to prolong the time taken to require dialysis [31] and improve the overall survival outcomes of CKD patients [32]. However, the outcome of long-term AST-120 use in CHD patients is unknown. Better nutrition clearly leads to higher serum IS and PCS levels. Therefore, further studies are needed to elucidate whether AST-120 has an add-on effect on the occurrence of cardiovascular events and the prognosis of dialysis patients with a good nutritional status.

The present study has several limitations. First, we only measured the total IS and PCS levels at the beginning of the study and then observed the occurrence of chosen events. We assumed that the serum levels of IS and PCS would not change over time or would at least increase by the same fraction in all patients. In future studies, a time-dependent Cox regression model [33] with multiple measurements of PCS and IS during the follow-up period will be considered. Second, this study was conducted at a single center, which may limit the generalizability of our findings. Finally, the sample size was small, and the number of outcome events was low; however, this concern may be trivial because a significant association was found between serum IS tertile and the occurrence of CV events. Moreover, we used the principal components analysis method to reduce the dimensionality, which is likely to have avoided over-adjusting in the multivariable analysis.

## 5. Conclusions

Serum IS levels correlated strongly with new onset of a CV event in our CHD patients, but an inverted U-curve association pattern was shown. Interestingly, a reverse epidemiology of IS on CVD was observed in CHD patients. However, further clinical studies are still required to confirm our findings.

## Figures and Tables

**Figure 1 jcm-10-00744-f001:**
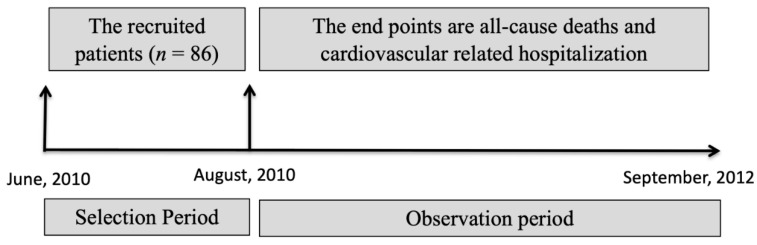
Study design.

**Figure 2 jcm-10-00744-f002:**
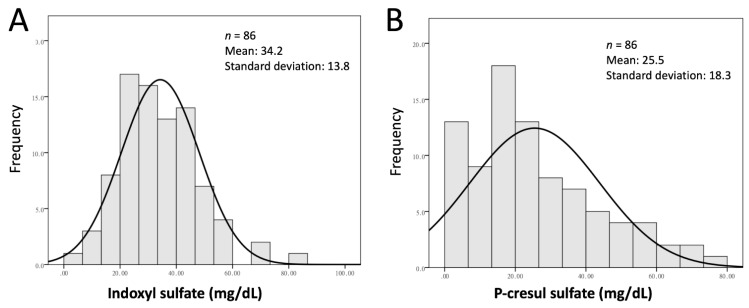
Distribution of serum levels of indoxyl sulfate (**A**) and P-cresyl sylfate (**B**).

**Figure 3 jcm-10-00744-f003:**
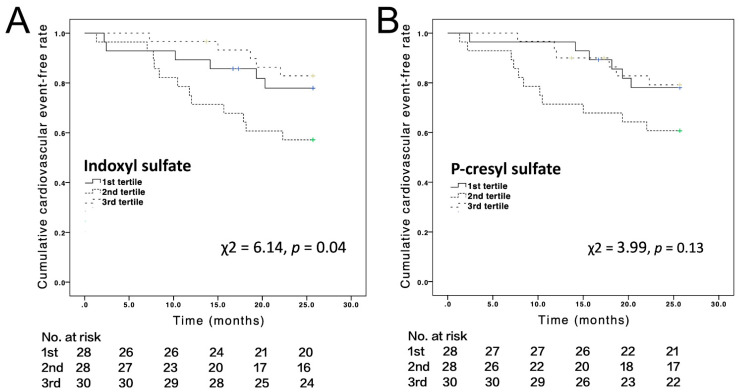
Kaplan–Meyer analysis of the associations of serum levels of (**A**) indoxyl sulfate and (**B**) P-cresyl sylfate with cardiovascular events.

**Figure 4 jcm-10-00744-f004:**
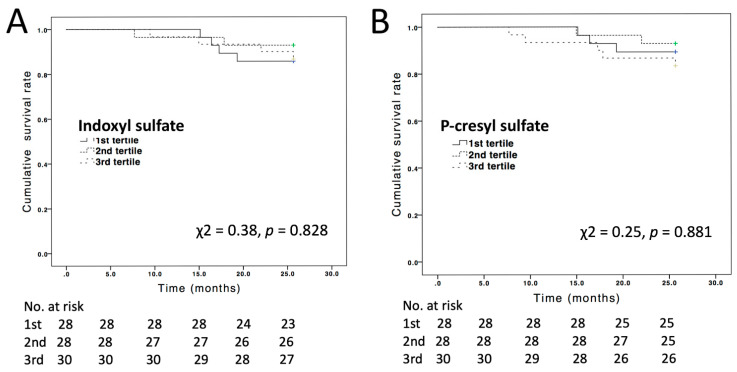
Kaplan–Meyer analysis of associations of serum levels of (**A**) indoxyl sulfate and (**B**) P-cresyl sylfate with survival.

**Table 1 jcm-10-00744-t001:** Clinical demographic and biochemical characteristics related to indoxyl sulfate and p-cresyl sulfate concentrations.

Parameters	All (*n* = 86)	Serum IS				Serum PCS			
		1st Tertile (*n* = 28)	2nd Tertile (*n* = 28)	3rd Tertile (*n* = 30)	*p* Value	1st Tertile (*n* = 28)	2nd Tertile (*n* = 28)	3rd Tertile (*n* = 30)	*p* Value
IS (mg/dL)	34.2 ± 13.8	5.5–26.2	26.6–39.1	39.5–81	-	-	-	-	-
PCS (mg/dL)	25.5 ± 13.8	-	-	-	-	0.5–15.5	15.7–28.	29.1–78.1	-
Age (year)	61.8 ± 12.4	63.0 ± 10.4	58.9 ± 14.6	63.3 ± 11.8	0.330	62 ± 12.3	59.3 ± 13.0	63.8 ± 11.8	0.388
Male gender (*n* %)	53 (61%)	17 (60%)	16 (57%)	20 (67%)	0.752	13 (46%)	20 (71%)	20 (67%)	0.123
Vintage (year)	9.1 ± 5.4	10.1 ± 5.8	8.9 ± 5.4	8.4 ± 5.0	0.591	10.5 ± 5.6	8.6 ± 5.0	8.3 ± 5.5	0.262
Body mass index (kg/m^2^)	22.4 ± 3.9	22.2 ± 2.7	22.3 ± 4.1	23.0 ± 4.8	0.867	22.6 ± 4.0	23.0 ± 4.2	22.0 ± 3.8	0.606
Diabetes mellitus (*n* %)	36 (42%)	12 (43%)	10 (35%)	14 (46%)	0.694	9 (32%)	12 (42%)	15 (50%)	0.384
Presence of CVD (*n* %)	26 (30%)	7 (25%)	10 (36%)	9 (30%)	0.683	7 (25%)	10 (36%)	9 (30%)	0.683
Systolic BP (mmHg)	148 ± 27	150 ± 26	145 ± 28	151 ± 27	0.600	146 ± 25	157 ± 28	145 ± 28	0.178
Diastolic BP (mmHg)	71 ± 17	71 ± 17	75 ± 16	69 ± 18	0.424	71 ± 11	77 ± 21	69 ± 18	0.204
Ca P product (mg^2^/dL^2^)	50.1 ± 15.5	40.6 ± 12.4	51.6 ± 18.3	52.1 ± 15.2	0.375	51.4 ± 16.1	52.2 ± 14.0	47.0 ± 16.2	0.271
iPTH (pg/mL)	186 ± 210	154 ± 250 ^a^	172 ± 179	229 ± 194	0.027	230 ± 274	182 ± 190	149 ± 149	0.349
Albumin (g/dL)	4.2 ± 0.4	4.1 ± 0.4 ^a^	4.1 ± 0.4 ^a^	4.4 ± 0.3	0.012	4.2 ± 0.4	4.3 ± 0.4	4.3 ± 0.5	0.737
hsCRP (mg/dL)	1.0 ± 1.9	0.8 ± 1.1	1.5 ± 3.0	0.9 ± 1.2	0.813	1.1 ± 1.2	0.9 ± 1.3	1.2 ± 2.8	0.846
Homocysteine (μmol/L)	26.6 ± 12.0	31.3 ± 17.0 ^a^	26.8 ± 7.6	22.2 ± 8.0	0.014	29.6 ± 17.3	25.4 ± 8.4	25.0 ± 8.4	0.284
Hemoglobin (g/dL)	10.4 ± 1.2	10.3 ± 1.2	10.4 ± 1.4	10.6 ± 1.3	0.639	10.2 ± 1.0	10.7 ± 1.6	10.3 ± 1.2	0.280
Ferritin (ng/mL)	566 ± 311	573 ± 329	481 ± 263	639 ± 328	0.084	651 ± 357	523 ± 286	525 ± 280	0.595
Uric acid (mg/dL)	7.0 ± 1.3	7.0 ± 1.3	7.2 ± 1.5	7.0 ± 1.2	0.675	7.2 ± 1.5	7.1 ± 1.1	6.8 ± 1.3	0.672
Total cholesterol (mg/dL)	184 ± 43	183 ± 47	174 ± 46	193 ± 35	0.119	181 ± 47	187 ± 45	183 ± 37	0.833
LDL (mg/dL)	109 ± 34	103 ± 32	105 ± 36	117 ± 33	0.275	97 ± 30	117 ± 37	111 ± 31	0.078
HDL (mg/dL)	49 ± 18	45 ± 14	45 ± 15	55 ± 22	0.105	50 ± 16	45 ± 21	51 ± 16	0.461
Triglycerides (mg/dL)	175 ± 215	229 ± 329	156 ± 157	141 ± 83	0.534	227 ± 356	166 ± 87	134 ± 75	0.249
Kt/V	1.38 ± 0.17	1.37 ± 0.13	1.40 ± 0.23	1.38 ± 0.17	0.970	1.40 ± 1.52	1.33 ± 1.64	1.40 ± 0.21	0.274

^a^*p* < 0.016, Mann–Whitney *U*-test vs. 3rd tertile. IS, indoxyl sulfate; PCS, p-cresyl sulfate; CVD, cardiovascular disease; BP, blood pressure; iPTH, intact parathyroid hormone; hsCRP; highly sensitive C-reactive protein; LDL, low-density lipoprotein; HDL, high-density lipoprotein; Kt/V, urea kinetics.

**Table 2 jcm-10-00744-t002:** Cox proportional regression model for evaluating the relationships among independent variables and clinical outcomes in patients with chronic dialysis.

Parameters	Cardiovascular Event		All-Cause Mortality	
	Crude		Crude	
	HR (95% CI)	*p*	HR (95% CI)	*p*
IS (per mg/dL)	1.00 (0.97–1.03)	0.899	1.00 (0.95–1.06)	0.96
1st tertile (vs. 3rd tertile)	1.39 (0.42–4.56)	0.585	1.63 (0.27–9.78)	0.591
2nd tertile (vs. 3rd tertile)	3.14 (1.10–8.94)	0.031	0.53 (0.14–7.51)	0.955
PCS (per mg/dL)	0.99 (0.97–1.02)	0.474	1.00 (0.97–1.04)	0.844
1st tertile (vs. 3rd tertile)	1.05 (0.34–3.27)	0.972	0.68 (0.11–4.11)	0.682
2nd tertile (vs. 3rd tertile)	2.31 (0.85–6.25)	0.099	0.67 (0.11–4.05)	0.669
Age (per year)	1.01 (0.97–1.04)	0.562	1.03 (0.96–1.10)	0.319
Male versus female	0.58 (0.26–1.33)	0.205	0.77 (0.17–3.47)	0.741
Vintage (per year)	1.02 (0.94–1.10)	0.554	1.11 (0.97–1.27)	0.115
BMI (per kg/m^2^)	0.97 (0.87–1.09)	0.689	0.94 (0.76–1.16)	0.587
Diabetes	0.68 (0.29–1.62)	0.395	0.21 (0.02–1.78)	0.154
Presence of CVD	2.64 (1.16–5.99)	0.020	0.36 (0.04–2.99)	0.345
Systolic BP (per mmHg)	0.99 (0.98–1.01)	0.488	0.99 (0.97–1.02)	0.892
Diastolic BP (per mmHg)	1.00 (0.97–1.02)	0.886	0.98 (0.94–1.03)	0.611
Ca P product (per mg^2^/dL^2^)	0.99 (0.96–1.02)	0.754	0.97 (0.91–1.03)	0.342
iPTH (per pg/mL)	0.99 (0.99–1.00)	0.473	0.99 (0.99–1.00)	0.287
Albumin (per g/dL)	0.29 (0.11–0.75)	0.011	0.05 (0.01–0.28)	0.001
hsCRP (per mg/dL)	1.26 (1.11–1.44)	<0.001	2.15 (1.41–3.30)	<0.001
Homocysteine (per μmol/L)	0.97 (0.93–1.01)	0.253	0.98 (0.91–1.06)	0.698
Hemoglobin (per g/dL)	0.92 (0.65–1.31)	0.675	0.52 (0.22–1.22)	0.135
Ln(Ferritin) (per 1 unit)	1.08 (0.69–1.70)	0.732	1.18 (0.48–2.89)	0.716
Uric acid (per mg/dL)	0.93 (0.68–1.25)	0.616	0.51 (0.28–0.90)	0.021
TC (per mg/dL)	0.99 (0.99–1.00)	0.351	0.97 (0.94–0.99)	0.027
LDL (per mg/dL)	0.98 (0.97–1.00)	0.022	0.97 (0.95–1.00)	0.060
HDL (per mg/dL)	1.01 (0.99–1.04)	0.116	1.01 (0.98–1.04)	0.509
Ln(Triglyceride) (per unit)	0.67 (0.36–1.24)	0.198	0.40 (0.12–1.35)	0.139
Kt/V (per unit)	4.69 (0.44–49.7)	0.199	4.64 (0.07–284)	0.464

Abbreviations: HR, hazard ratio; CI, confidence interval; IS, indoxyl sulfate; PCS, p-cresyl sulfate; BMI, body mass index; CVD, cardiovascular disease; BP, blood pressure; iPTH, intact parathyroid hormone; hsCRP; Ln, natural log transformation; high sensitive C-reactive protein; TC, total cholesterol; LDL, low-density lipoprotein; HDL, high-density lipoprotein; Kt/V, urea kinetics.

**Table 3 jcm-10-00744-t003:** Stepwise multivariate Cox regression analysis of risk factors at baseline for experiencing a cardiovascular event: indoxyl sulfate concentrations shown as tertiles.

Models	HR	95% CI	*p* Value
Unadjusted			
1st tertile	1.39	0.42–4.56	0.58
2nd tertile	3.14	1.10–8.94	0.03
3rd tertile	1		
Model 1			
1st tertile	1.35	0.41–4.46	0.61
2nd tertile	3.70	1.25–10.92	0.01
3rd tertile	1		
Model 2			
1st tertile	1.76	0.52–5.97	0.36
2nd tertile	4.27	1.37–13.22	0.01
3rd tertile	1		
Model 3			
1st tertile	2.25	0.67–8.04	0.214
2nd tertile	4.63	1.40–15.37	0.012
3rd tertile	1		
Model 4			
1st tertile	1.76	0.43–7.19	0.430
2nd tertile	3.31	0.87–12.60	0.079
3rd tertile	1		
Model 5 *			
1st tertile	2.19	0.63–7.64	0.218
2nd tertile	5.42	1.67–17.60	0.005
3rd tertile	1		

Model 1 is adjusted for age, sex, and hemodialysis time. Model 2 comprises model 1, as well as adjustments for diabetes mellitus and previous cardiovascular disease. Model 3 comprises model 2, as well as adjustments for homocysteine, calcium phosphate product, and highly sensitive C-reaction protein concentrations. Model 4 comprises model 3, as well as low-density lipoprotein and albumin concentrations. Model 5 comprises model 2 and two principal components. * The two principal factors used were dimension reduction from the parameters of homocysteine, ionized calcium, phosphate, sensitive C-reaction protein, low-density lipoprotein, high-density lipoprotein, albumin, triglycerides, ferritin, uric acid, body mass index, and hemoglobin. Abbreviations: HR, hazard ratio; CI, confidence interval.

## Data Availability

The data presented in this study are available in Appendix A.

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
