# Peer review of "Inverted U-Curve Association between Serum Indoxyl Sulfate Levels and Cardiovascular Events in Patients on Chronic Hemodialysis"

_jcm, 2021, doi:10.3390/jcm10040744_

Round 1

Reviewer 1 Report

The authors investigated the relationship between serum indoxyl sulfate (IS) levels and cardiovascular events in patients who had undergone chronic hemodialysis. They showed that the presence of inverse U-curve relationship between them. The point of view of present study seems to be interesting, however, there are several problems to be solved.

#1   In the present study, there were 23 cardiovascular events among 86 studied patients. In the Cox proportional hazard model, there were many factors included in the present analyses. In general, the factors for such analyses are considered as appropriate when the numbers of factors x 10 = the numbers of cardiovascular events. The authors should consult with statistician to confirm the number of factors for analyses and your results appropriately.                                                                                                                                             #2   The authors speculated that a higher albumin overcome the cardiovascular risks induced by IS, in the "Discussion" section. However, this speculation did not seem to be accepted easily. From the start, was the hypothesis that an increasing IS can affect the development of cardiovascular events, supported by obvious evidences? First, the authors should provide such data sufficiently and, then, should add another speculations for an inverse U-curve relationship between IS and cardiovascular events.                                                                                                                                                                                #3 In the "Abstract" section, the number of cardiovascular events was not shown. The authors should add the number of cardiovascular events in such area.                                                                                                                                                                                                                                #4 Regarding the kinds of cardiovascular events, atrial fibrillation was included in it, however, the occurrence of atrial fibrillation was due to the increasing age, the authors had better exclude this factor from the cardiovascular events.

Reviewer 2 Report

It's a new cohort of hemodialysed patients studiying impact of uremic toxins on mortality and CV events. 

Their results are interesting and provide arguments on cardiovascular toxicity of uremic toxins

Comments (minor revision):

1) On introduction, when the authors talk about pathological effect of indoxyl sulfate, it's necessary they talk about the endothelial toxicity of indoxyl sufate, that is a principal pathophysiological process in cardiovascular toxicity of this toxin (and cite DOI: 10.3390/toxins12040229)

2) Why the authors chose the level of 2.9g/dl for serum albumine for excluded patients with poor nutrition status? and how many patients does that represent?

3) How many  patients were excluded in total? we don't found the response in the results

4) The figure 4 is not pretty, and too small. It's necessary to give the significativity (p) for each of the kaplan meyer analysis in the figure

Thank you

Round 2

Reviewer 1 Report

I have no request and comment regarding the revised manuscript.